# Use of Remdesivir in Patients Hospitalized for COVID-19 Pneumonia: Effect on the Hypoxic and Inflammatory State

**DOI:** 10.3390/v15102101

**Published:** 2023-10-17

**Authors:** Alessandro Libra, Nicola Ciancio, Gianluca Sambataro, Enrico Sciacca, Giuseppe Muscato, Andrea Marino, Carlo Vancheri, Lucia Spicuzza

**Affiliations:** 1Regional Referral Centre for Rare Lung Disease, University Hospital “Policlinico-San Marco”, Department of Clinical and Experimental Medicine, University of Catania, 95123 Catania, Italy; nicolaciancio@hotmail.com (N.C.); dottorsambataro@gmail.com (G.S.); esciacca29@gmail.com (E.S.); gpp.muscato@gmail.com (G.M.); vancheri@unict.it (C.V.); lucia.spicuzza@unict.it (L.S.); 2Department of Biomedical and Biotechnological Sciences, Unit of Infectious Diseases, University of Catania, 95123 Catania, Italy; andrea.marino@unict.it

**Keywords:** remdesivir, respiratory failure, antiviral therapy, IL6, SARS-CoV-2, P/F ratio

## Abstract

Remdesivir is one of the most attractive options for patients with hypoxemic respiratory failure due to coronavirus disease 2019 (COVID-19). The aim of our study was to evaluate the effect of remdesivir on the hypoxic and inflammatory state in patients with moderate to severe COVID-19. We retrospectively enrolled 112 patients admitted for COVID-19 pneumonia, requiring low-flow oxygen, 57 treated with remdesivir plus standard of care (SoC) and 55 treated only with SoC that were similar for demographic and clinical data. We evaluated changes in hypoxemia and inflammatory markers at admission (Day 0) and after 5 days of treatment (Day 5) and the clinical course of the disease. From Day 0 to Day 5, the ratio of arterial oxygen partial pressure to fractional inspired oxygen (P/F) increased from 222 ± 62 to 274 ± 97 (*p* < 0.0001) in the remdesivir group and decreased from 223 ± 62 to 183 ± 76 (*p* < 0.05) in the SoC group. Interleukine-6 levels decreased in the remdesivir (45.9 to 17.5 pg/mL, *p* < 0.05) but not in the SoC group. Remdesivir reduced the need for ventilatory support and the length of hospitalization. In conclusion, compared to standard care, remdesivir rapidly improves hypoxia and inflammation, causing a better course of the disease in moderate to severe COVID-19.

## 1. Introduction

The coronavirus disease 2019 (COVID-19) pandemic, due to the severe acute respiratory syndrome coronavirus 2 (SARS-CoV-2), has caused a worldwide sudden and substantial increase in hospitalizations for pneumonia and multi-organ disease as well as in mortality [1]. Hospitalization has been common, particularly during the first two waves of the pandemic, in the pre-vaccine era, when the virus was the cause of severe and fatal pneumonia. SARS-CoV-2-related pneumonia is associated with hypoxemic respiratory failure, the severity of which varies across different populations and through the different waves of the pandemic [2]. At this time, when the emergency phase has just passed, older age, chronic diseases and oncological conditions, particularly hematological, still represent a major risk factor for COVID-19-related severe respiratory failure and intensive care requirement [3]. Indeed, proper management of patients with COVID-19 includes best practices for supportive care of acute hypoxic respiratory failure; however, since the onset of the pandemic, intense research has been conducted in order to identify specific pharmacological treatments [4]. Among different antiviral agents taken into consideration, remdesivir (GS-5734), an inhibitor of the viral RNA-dependent RNA polymerase with in vitro inhibitory activity against SARS-CoV-1 and Middle East Respiratory Syndrome-CoV, raised an early interest for its ability to inhibit SARS-CoV-2 in vitro [5]. In the last years, several studies investigated the clinical effects of remdesivir in patients with COVID-19, with conflicting results [6,7,8,9,10]. Some early randomized controlled trials (RCTs) showed that the rate of clinical improvement was significantly higher in groups of hospitalized patients with COVID-19 treated for 5 or 10 days with remdesivir compared to control groups treated with standard care [6,7]. Though, other RCTs failed to find a difference in the clinical status of patients randomized to remdesivir or to standard care within the first 5–10 days from treatment initiation [8,11,12]. Overall, these studies included different populations in terms of severity of the disease, concomitant treatments and timing of treatment initiation and various outcomes were evaluated [9]. Some recent studies suggest that remdesivir may improve the clinical course of the disease in selected patients, such as those that are noncritical or requiring low flow of oxygen, although data on mortality still remain inconclusive [9,10,13,14]. Most of the available studies on remdesivir focused on the clinical course of the disease evaluated in terms of mortality, days of hospitalization, scoring at a variety of clinical scales, level of required respiratory support and admission to Intensive Care Unit (ICU) [9].

From the early stages of the COVID-19 pandemic, it has been recognized that, from a physiologic standpoint, the degree of hypoxemia and the degree of the inflammatory response, mainly associated with the so-called “cytokines storm” are major factors affecting outcomes in patients with COVID-19 pneumonia [15,16].

Therefore, the present study was designed to assess the efficacy of remdesivir in improving the hypoxic state and the inflammatory response in patients hospitalized for COVID-19 pneumonia and hypoxic respiratory failure.

The PaO_2_/FiO_2_ (P/F) is the ratio of arterial oxygen partial pressure (mmHg of PaO_2_) to fractional inspired oxygen (FiO_2_) and is widely used in clinical practice as an indicator of the severity of the hypoxemic respiratory failure for its accuracy and simplicity [17,18]. The P/F, first utilized to assess patients with acute respiratory distress syndrome (ARDS), has been proposed as a prognostic marker in patients with COVID-19 [15]. For this reason, we decided to use this as our primary outcome also since, to the best of our knowledge, it has not been investigated before.

## 2. Materials and Methods

### 2.1. Study Design and Patients’ Population

This is a mono-centric, observational, retrospective study, comparing outcomes in patients hospitalized for COVID-19-related pneumonia treated with remdesivir plus standard of care (study group) or with standard of care (SoC group). This study was conducted in the Respiratory Care Unit of “Azienda Ospedaliera Universitaria P.O. G. Rodolico—S- Marco”, a primary care unit designated as a COVID-19 Unit. This research was conducted according to the Declaration of Helsinki. It was approved as a retrospective minimally invasive experimental study by the Provincial Review Board of Messina on 29 June 2020, with the protocol number 63/20 bis. Patients signed written informed consent for the use of their data for research purposes at admission.

We retrospectively analyzed data from patients hospitalized during the first and second waves of the SARS-CoV-2 pandemic, from June 2020 to May 2021.

The use of remdesivir for COVID-19 was approved in Italy on 26 November 2020. According to the Italian Agency for Pharmaceutics (AIFA), to be eligible for treatment with remdesivir, patients >18 years old had to meet the following criteria: (1) hospitalization for SARS-CoV-2 infection documented by molecular testing; (2) pneumonia documented by any kind of imaging; 3) symptom onset from less than 10 days; (3) need for supplemental oxygen (SpO2 < 92% at room air or P/F < 300; (4) no need for noninvasive mechanical ventilation (NIV), invasive mechanical ventilation (IMV) or extracorporeal membrane oxygenation (ECMO); (5) estimated glomerular filtration rate (eGFR) value > 30; (6) alanine aminotransferase (ALT) within normal values or <5 times the upper limit of normal range at baseline; and (7) normal values of conjugated bilirubin, alkaline phosphatase and normal prothrombin time.

In our unit, after November 2020, treatment with remdesivir was offered to all patients meeting these criteria. In the remdesivir group, we included all patients >18 years, who were eligible for treatment and accepted remdesivir in addition to standard care. In order to establish the composition of the control SoC group, we screened patients >18 years hospitalized for COVID-19 pneumonia prior to November 2020 or patients who refused remdesivir, who required oxygen treatment but not NIV or IMV. From a first analysis, we found that, without any selection, on admission, the two groups were identical for demographic, clinical history and laboratory parameters. The only exception was that the group treated with SoC included 3 patients >90 years old that we excluded from the group. We also excluded from both groups pregnant or lactating women.

### 2.2. Treatment Protocols

According to a standard protocol, the dose regimen of remdesivir was 200 mg IV, infused over 30 to 60 min on day 1, followed by 100 mg for the next four days. All patients received supportive care according to standard of care. The SoC consisted of dexamethasone 6 mg IV and enoxaparin sodium at prophylactic dosage or according to clinical decision. Antibiotics were used when there was clinical or laboratoristic suspicion of infection. Supplemental oxygen was given in order to maintain oxygen saturation (SaO_2_) between 92% and 96%.

### 2.3. Patients’ Management and Data Collection

Data were obtained by reviewing medical records containing demographic data, clinical history, exposure history, laboratory, clinical and radiological data. Upon admission, an anamnesis was collected aimed at recognizing chronic diseases and the relevance of co-morbidities was assessed using the Charlson Index [18]. Patients were successively evaluated for symptoms from onset to admission (fever, cough and dyspnea) and vital signs were recorded. Arterial blood sample was obtained from gas analyses to evaluate pH, PaO_2_, PaCO_2_ and the P/F ratio. Routine blood chemistry was performed, including, in addition to standard inflammation markers, interleukin 6 (IL-6) and ferritin. Pneumonia was confirmed in all patients by computed tomography (CT). During hospital stay, patients were evaluated every day by an expert pneumologist and the P/F ratio and vital signs were daily recorded. Day 0 refers to admission. In the group treated with remdesivir, the treatment was administered from Day 1 to Day 5. For each patient included in the study, we collected the P/F ratio, PaCO_2_ and laboratory data at Day 0 and at Day 5. From clinical records, we also assessed the length of hospitalization, the rate of patients who worsened during hospitalization requiring an escalation of respiratory support and patients who were admitted to Intensive Care Unit (ICU) or died during hospitalization. Discharge criteria were two consecutive negative PCR tests; no fever for at least 2 days; and improvement in gas exchange.

### 2.4. Outcomes

Our primary outcomes were changes induced by remdesivir in P/F ratio and inflammatory markers from admission to Day 5. Secondary outcomes were the need for higher level of respiratory support during the hospitalization period and the length of hospitalization, reflecting the clinical course of the disease.

### 2.5. Statistical Analysis

Descriptive data are presented as mean ± standard deviation (SD). Comparison among groups was performed by unpaired Student’s *t*-test, while a paired *t*-test was used to compare changes from Day 0 to Day 5. Chi square tests were used to compare nominal data. Correlations between variables were made with Pearson’s test for parametric variables and Spearman’s test for nonparametric variables. Statistical analysis was performed using GraphPad Prism Version 9.3.

## 3. Results

### 3.1. Characteristics of the Study Groups

We reviewed a total of 134 clinical records from patients admitted for COVID-19 pneumonia. Among these, 112 met the inclusion criteria and were selected for the study. Of these 112 patients, 55 were treated only with SoC and 57 received remdesivir in addition to SoC. No patient in the two groups had been vaccinated for SARS-CoV-2. According to the standard WHO definition, our patients had moderate to severe COVID-19 as all required oxygen support [19]. Demographic, clinical characteristics, laboratory and respiratory data of the patients at baseline are shown in Table 1. The two groups were similar for sex distribution (>2/3 males), mean age (61 years remdesivir vs. 63 years SoC, P = NS) and body mass index (BMI) media was 28 in both groups. The burden of comorbidities was also similar in the two groups (Charlson Index 2). Laboratory values were also similar. Both groups exhibited abnormal high values of IL-6, C-reactive protein (CRP), lactate dehydrogenase (LDH) and ferritin. Also, D-dimer levels were higher than normal range in both groups (Table 1). The P/F ratio (PaO_2_ in mmHg) at admission was 222 (range 91–296) in the remdesivir group and 223 (range 92–298) in the SoC group, respectively (Table 1). We found that, in the total enrolled population, the P/F ratio at baseline weakly but significantly inversely correlated with IL-6 values (r = −0.27, *p* < 0.05) but not with CRP values.

### 3.2. Effect of the Treatment with Remdesivir on the Hypoxic State and Disease Course

In the group treated with remdesivir plus SoC, we observed an improvement in the P/F ratio from 222 ± 62 at Day 0 to 274 ± 97 at Day 5 (*p* < 0.0001). Conversely, in the group treated with SoC, there was a significant decrease from 223 ± 62 at Day 0 to 183 ± 76 (*p* < 0.05) at Day 5 (Table 2, Figure 1). Escalation of respiratory support was defined as the need for NIV or IMV. An increase in the respiratory support was necessary for 15 patients (26.3%) in the group treated with remdesivir plus SoC and 28 in the group treated with SoC alone (49.1%) (*p* < 0.05). In addition, the mean duration of the hospitalization was shorter in the remdesivir group compared SoC group (15.3 ± 8.3 vs. 20.1 ± 8.8 days, respectively, *p* < 0.05) (Figure 2). We found an inverse correlation between the absolute change in P/F ratio values from Day 0 to Day 5 and the days of hospitalization (r = −0.44, *p* < 0.0001) (Figure 3). Only a small number of patients needed admission to ICU for IMV (one in the remdesivir plus SoC group and two in the SoC group) and only one patient in the SoC group died during hospitalization (Table 3). Treatment with remdesivir was well tolerated by all patients. Kidney function remained normal. Liver enzymes slightly increased in both groups, although the increase did not reach statistical significance in the SoC group (Table 2).

### 3.3. Effect of the Treatment with Remdesivir on the Inflammatory State

In both groups at baseline, serum levels of inflammatory markers were higher than normal ranges (Table 1). A significant reduction in CRP values was observed from baseline to Day 5, from 87 to 13 mg/L in the remdesivir plus SoC group and from 90 to 33 mg/L in the SoC group (Table 2).

In contrast, IL-6 values were reduced in the remdesivir plus SoC group from 45.9 pg/mL at Day 0 to 17.5 pg/mL at Day 5 (*p* < 0.05), whereas an increase in values was observed in the group treated with SoC (Figure 1). In the whole cohort at Day 5, the correlation between IL-6 values and P/F ratio was still significant (r = −0.28, *p* < 0.05). Also, serum ferritin was significantly reduced, from 624 ng/mL to 483 ng/mL after 5 days in the remdesivir but not in the control group.

## 4. Discussion

This real-life retrospective case–control study highlights some specific beneficial effects of treatment with remdesivir in association with standard supportive care, compared to standard supportive care alone, in patients hospitalized with hypoxic respiratory failure from moderate to severe COVID-19, requiring oxygen therapy but not NIV or IMV. These effects include an early improvement in the hypoxic state, associated with a reduction in the inflammatory response. Moreover, remdesivir improved the course of the disease, as indicated by a lower rate of respiratory support escalation and a shorter length of hospitalization. As death occurred only in a few patients, no data could be obtained on mortality.

Since the start of the COVID-19 pandemic, an overwhelming amount of research has been performed in order to find an effective pharmacological treatment, unfortunately with inconclusive results, and prevention with vaccination remains the key approach to control the pandemic [20]. For the treatment of hospitalized patients, in addition to respiratory support and anti-inflammatories, antivirals are the primary therapies used in clinical practice and the object of ongoing investigations.

At the time we are writing, most nations have declared concluding the emerging phase of the pandemic and it has been suggested that the disease will become endemic and will be characterized by a lower pathogenicity [21]; although the incidence of SARS-CoV-2-related pneumonia has dramatically fallen, given a large part of the world population vaccinated, still it represents a major threat in some specific populations, including nonvaccinated individuals, elderly, patients with chronic conditions and oncologic patients, particularly those with hemato-oncological disease [3,22]. Therefore, the search for effective drugs still remains crucial.

A rapid improvement in hypoxic condition after treatment with remdesivir is the main novelty of our study. As far as we know, this aspect has not been previously addressed, as, in most of the cases, the clinical course of the disease has been evaluated through clinical progression scales (mainly the seven-point WHO scale). Indeed, hypoxemia at admission is a major risk factor for prognosis in patients with COVID-19-associated pneumonia [15,16]. Often in these patients, low PaO_2_ values are associated with acceptable values of SaO_2_, thus leading to an underestimation of the severity of respiratory impairment [16]. Evaluation of the P/F ratio provides an easy and valuable approach to understand the severity of the condition at admission and during follow-up [15,16]. This indicator has already been used to guide clinical decisions in ARDS (or requirement of ICU) and has been considered a reliable tool to stratify the mortality risk [17].

Likewise, in patients with acute hypoxemic failure from COVID-19, early studies showed that the P/F ratio was one of the most reliable parameters to identify those who were at risk of developing severe respiratory failure, to predict in-hospital mortality and the need for more aggressive supportive therapies [15,16,23]. A moderate to severe impairment in P/F ratio has been independently associated with a threefold increase in the risk of in-hospital mortality [16]. In particular, Sinatti and colleagues found that, in SARS-CoV-2-infected patients, a P/F ratio <274 mmHg was a good predictive index test to forecast the development of severe respiratory failure [15]. Reasonably, a rapid improvement in the P/F ratio is expected to be associated with improved prognosis. Indeed, we found in our cohort a significant correlation between the absolute change in P/F ratio from Day 0 to Day 5 and the length of hospitalization.

It is difficult to make a comparison between our findings and other studies, as the P/F has hardly been considered, among other clinical indicators. The NOR-Solidarity Trial is an independent add-on study to the early WHO Solidarity Trial [24]. This study, was designed to evaluate the effect of remdesivir on more specific outcomes such as the viral clearance, the degree of respiratory failure and inflammatory variables. It was reported that, in a small group of patients with a mean P/F ratio of 38 kPa (285 mmHg), which is higher compared to our cohort, remdesivir caused a slight improvement (around 45 kPa, 337 mmHg) at Day 7. However, in accordance with the WHO Solidarity Trial, no reduction in the rate of mortality, ICU admission or escalation of respiratory support was reported [24]. One study including patients with severe COVID-19 requiring treatment with NIV, reported that remdesivir increased the P/F ratio from 101 to 204 by the fifth day of treatment; however, the study lacked a control group [25].

In another study including patients with severe respiratory failure, the basal P/F ratio significantly increased from 120 to 280 in a group treated with remdesivir plus tocilizumab; however, the effect of remdesivir alone or standard treatment alone was not investigated [26].

We found that, in patients in the group treated with remdesivir, the risk to require a higher level of respiratory support was nearly half compared to patients treated with standard care alone. This is not surprising considering that, in this group, hypoxemia rapidly improved, while a worsening was observed in patients treated with standard care. Previous studies exploring this aspect have given inconsistent results [11,24]. The WHO as well the NOR-Solidarity trials failed to show that remdesivir reduced the need for mechanical ventilation during hospitalization [11,24]. However, in a meta-analysis including five RCTs, Rezagholizadeh and colleagues showed that remdesivir administration was associated with a significant reduction in IMV or ECMO requirement through days 14–28 [27].

A more recent meta-analysis focused on patients with COVID-19 requiring oxygen treatment [28]. Remdesivir lowered the risk of progression to NIV among patients on any supplemental oxygen (RR: 0.56) and low-flow oxygen (RR: 0.37) and also lowered the risk of progression to invasive mechanical ventilation with similar risk ratios. For both NIV and IMV, treatment with remdesivir was ranked superior to the standard of care across all patient subgroups [28]. In our study, the beneficial effect of remdesivir was also confirmed by the shorter length of hospitalization in the study group and this is in accordance with previous data [9]. We also found that the level of improvement in hypoxemia during the first 5 days correlated with the duration of hospitalization, confirming the predictive role of P/F at admission on the course of the disease and the beneficial effect of remdesivir in this early phase.

The high levels of blood inflammatory markers that we found reflect systemic inflammation associated with COVID-19 [1,29]. It is well acquired that IL-6, a prototype cytokine with pleiotropic activity, is one of the primary agents involved in the inflammatory response to infection caused by viruses, including SARS-CoV-2 [30,31]. As such, during the COVID-19 pandemic, the role of IL-6 has been widely explored as a diagnostic tool, to assess early inflammation and the risk of sepsis, and to address therapeutic strategies [1]. A high level of serum IL-6 is an indicator of cytokine release syndrome (CRS), consistently reported by several studies on COVID-19 and associated with the severity of the disease, such that some authors consider IL-6 an irreplaceable marker of CRS [1,30].

More recently, high levels of IL-6 have been shown in long-COVID [31]. We found that IL-6 levels were associated with the degree of hypoxia. The correlation was weak, although significant, perhaps due the smallness of the sample. Interestingly remdesivir markedly reduced IL-6 levels that remained unchanged in the control group. This finding may be relevant as the inhibition of inflammatory response has been considered an important approach to prevent the cytokine release syndrome toxicity in COVID-19 so that some attempts have been made to selectively inhibit IL-6 after virus infection [32,33].

Although in vitro studies showed that remdesivir reduced IL-6 production induced by human coronavirus OC43 in human lung fibroblasts [34], little is available on the effect of remdesivir on the inflammatory response in clinical studies on COVID-19. In the NOR-Solidarity randomized trial, the effect of remdesivir on the degree of inflammation could not be consistently proved, as CRP basal values were different in the study and control groups and IL-6 levels were not evaluated [24]. Conversely, in the study by Simioli et al. on severe patients with COVID-19, IL-6 was significantly reduced within 6 days of treatment with remdesivir; however, this study had no control group [25].

More recently one study explored the immunomodulatory effect induced by remdesivir in patients with critical COVID-19 admitted to ICU. After 5 days of treatment, remdesivir significantly reduced serum levels of inflammatory Th1-type and Th17-type inflammatory cytokines and increased Th2-type cytokines [35]. The exact mechanism underlying this modulatory effect of remdesivir remains to be clarified; however, it has been speculated that, if hyperactivation of immune cells driving secretion of inflammatory cytokines may accelerate tissue remodeling and contribute to COVID-19 progression, this aspect may be beneficial [35].

Interestingly, we also found that remdesivir, but not standard care, reduced serum levels of ferritin, known to be higher in patients with COVID-19 [36]. Although iron parameters are not regarded as biomarkers predicting septic development and are not mentioned in the recommendations for the management of COVID-19 the relationship between IL-6 and iron metabolism is well recognized [36,37].

How these physiological effects of remdesivir translate into an improvement in clinical outcomes is, so far, uncertain and probably the choice of the right patient and right time to administer remdesivir is fundamental.

At the time we are writing, a number of systematic reviews and meta-analyses have tried to draw conclusions, with conflicting results [9,27,28,34,38,39]. Some of the most recent meta-analyses agree that treatment with remdesivir, compared to supportive care, may improve clinical recovery, reduce adverse events, reduce the risk of invasive ventilation and intrahospital mortality in non-severe patients hospitalized for COVID-19 [27,28,38,39,40]. However, some meta-analyses have not confirmed benefits on mortality [9,38,40]. The most recent Cochrane review on remdesivir, including RCTs on remdesivir vs. standard care, concludes that, in individuals with moderate to severe COVID-19, remdesivir may be beneficial in the clinical course of the disease but probably makes little or no difference to all-cause mortality at up to day 28 (certainty of the evidence remains low to moderate) [9].

It is likely that the uncertain effect of remdesivir on mortality is due to the fact that patients often are not stratified by risk. Interestingly, a large study on over 18,000 patients failed to show an effect of remdesivir on overall mortality, although a significant reduction was observed in patients with low-flow oxygen requirement [13]. These data are also confirmed in the meta-analyses by Lee et al. showing that remdesivir reduce the risk of mortality, compared to standard care, in nonventilated patients requiring oxygen treatment [40]. In addition, more recently, it has been shown that remdesivir reduces mortality in patients affected by hemato-oncological conditions, still representing a major risk factor for COVID-19 and preliminary data suggest that may be beneficial in patients with lymphoma showing prolonged symptoms and viral shedding [3,41,42].

Finally, remdesivir was safe in our patients. A slight elevation in aminotransferase was observed in both groups, although statistical significance was reached only for the treatment group. Although it has been described that remdesivir can cause hepatocellular injury, it is also possible that SARS-CoV-2 infection may increase aminotransferase levels [43]. However, in our study, the number of patients included was insufficient to address safety issues.

This study has strengths and limitations. One strength is that the study was conducted in a real-world clinical setting; thus, results may be applicable to other similar populations. However, this was not a randomized trial, although the two groups were perfectly matched. In addition, due to the small sample, we were not able to draw a conclusion on intrahospital mortality. The present study was conducted during the early stages of the COVID-19 pandemic when there were no available oral antiviral treatments or vaccines. This context is crucial as it might affect the generalizability of our findings to later stages of the pandemic when such treatments and preventive measures became available. It is essential to interpret the results with this in mind, recognizing that the conditions and available interventions during our study period were different from later stages of the pandemic.

## 5. Conclusions

In conclusion, as SARS-CoV-2 is now becoming endemic, pneumonia still remains a major issue for a large number of patients affected by a variety of chronic conditions or in a state of immunosuppression. In patients with moderate–severe COVID-19, remdesivir, compared to standard care alone, rapidly improves hypoxemia and attenuates the inflammatory response, thus improving the course of the disease and avoiding progression.

## Figures and Tables

**Figure 1 viruses-15-02101-f001:**
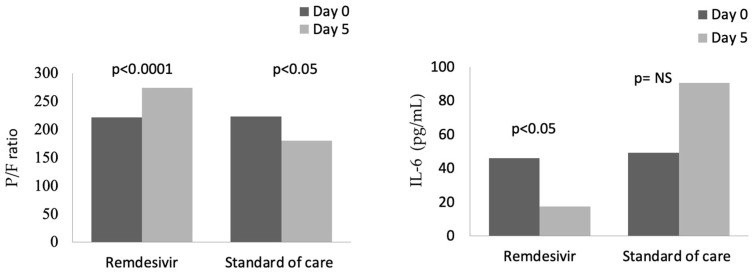
Changes in P/F and IL-6 from baseline (Day 0) to Day 5 in patients treated with remdesivir or standard of care. NS: not significant.

**Figure 2 viruses-15-02101-f002:**
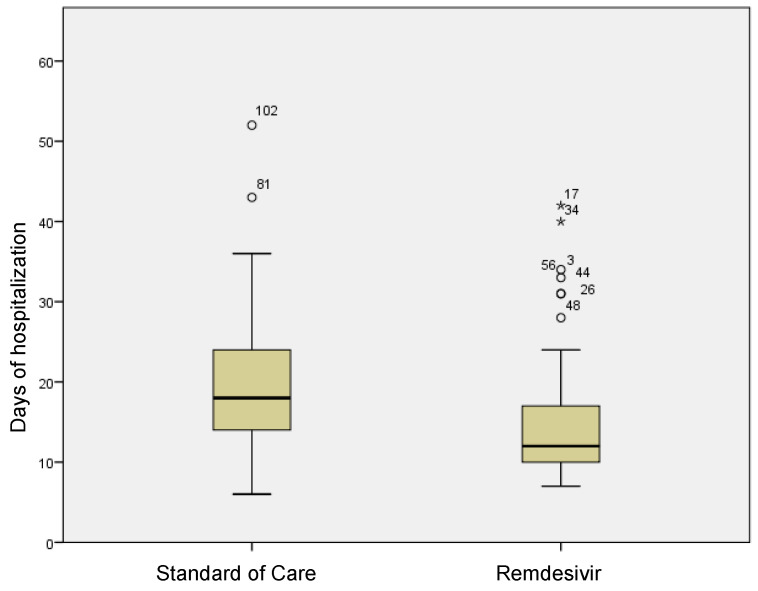
Differences in terms of length of stay in remdesivir and standard of care patients. Stars in the figure shows extreme outlier.

**Figure 3 viruses-15-02101-f003:**
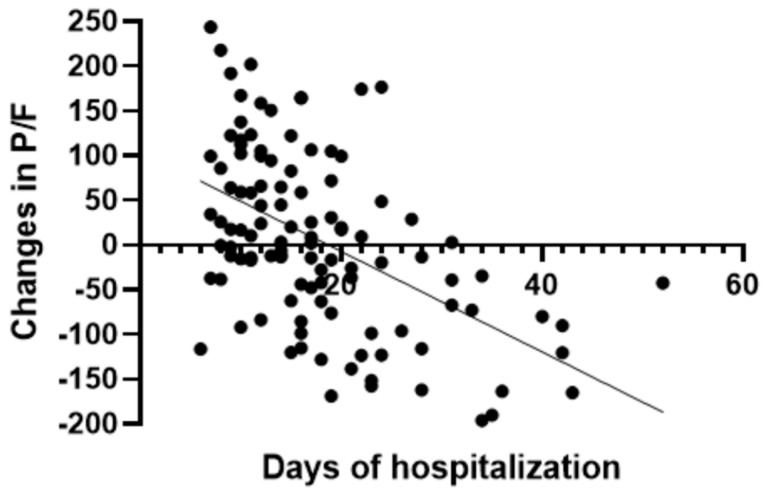
Correlation between absolute change in P/F ratio and days of hospitalization in all patients.

**Table 1 viruses-15-02101-t001:** Demographic and clinical data of patients with COVID-19 at admission.

Item	Remdesivir (57)	Standard of Care (55)
Male sex (%)	69.1%	78.9%
Age (years)	61.2 ± 10.8	63.4 ± 11.5
Charlson Index Score	2.4 ± 1.3	2.7 ±1.8
BMI	28.1 ± 5.4	28.2 ± 2.8
P/F (mmHg)	222 ± 62	223 ± 62
PCO_2_ (mmHg)	35.2 ± 6.5	34.2 ± 4.9
RBC (×10^6^/μL)	4.6 ± 0.5	4.7 ± 0.7
Hb (g/dL)	13.8 ± 1.2	13.2 ± 2.2
WBC (×10^3^/μL)	7.7 ± 3.4	8.2 ± 4.1
PLT (×10^3^/μL)	242 ± 84	235 ± 94
Neu (×10^3^/μL)	6.1 ± 3.4	6.7 ± 3.9
Lym (×10^3^/μL)	1 ± 0.4	1.1 ± 0.6
Mon (×10^3^/μL)	0.3 ± 0.5	0.3 ± 0.4
Eos (×10^3^/μL)	0 ± 0	0 ± 0
Bas (×10^3^/μL)	0.0 ± 0.2	0 ± 0
CRP (mg/L)	87.2 ± 61.3	90.2 ± 60.5
Ferritin (ng/mL)	624 ± 460	613 ± 426
IL6 (pg/mL)	45.9 ± 59.5	48.5 ± 48.7
eGFR (mL/min/1.73 m^2^)	86.5 ± 17.1	82.7 ± 25.9
D-Dimer (μg/L)	325 ± 292	389 ± 500
AST (U/L)	39.1 ± 19	34 ± 19.7
ALT (U/L)	38.2 ± 22.6	39 ± 48.8
LDH (U/L)	332 ± 114	336 ± 105
PCT (ng/mL)	0.1 ± 0.5	0.1 ± 0.4

Data expressed as mean ± standard deviation; *p* values were not significant for all data. BMI: body mass index in kg/m^2^; PF: PaO_2_/FiO_2_; pCO_2_: blood partial pressure of CO_2_; RBC: red blood cells; Hb: hemoglobin; WBC: white blood cells, PLT: platelets; Neu: neutrophils; Lym: lymphocytes; Mon: monocytes; Eos: eosinophils; Bas: basophil; CRP: C-reactive protein; IL-6: interleukin 6; eGFR: estimated glomerular filtration rate; AST: aspartate transaminase; ALT: alanine transaminase; LDH: lactate dehydrogenase; PCT: procalcitonin.

**Table 2 viruses-15-02101-t002:** Differences in characteristics of COVID-19 patients on Day 0 and Day 5, after treatment with remdesivir plus SoC or SoC alone.

	Remdesivir			Standard of Care		
	Day 0	Day 5	*p*-Value	Day 0	Day 5	*p*-Value
P/F (mmHg)	222 ± 62	274 ± 97.4	<0.0001	223 ± 62	183 ± 76	0.02
RBC (×10^6^/μL)	4.6 ± 0.5	4.5 ± 0.6	0.28	4.7 ± 0.7	4.5 ± 0.7	0.09
PLT (×10^3^/μL)	242 ± 84.8	357 ± 105	<0.0001	235 ± 94.7	313 ± 106	<0.0001
Hb (g/dL)	13.8 ± 1.2	15.4 ± 1.3	0.44	13.2 ± 2.1	12.6 ± 1.7	<0.0001
WBC (×10^3^/μL)	7.7 ± 3.4	9.6 ± 3.1	<0.0001	8.2 ± 4.1	8.6 ± 3.1	0.4
Neu (×10^3^/μL)	6.1 ± 3.4	6.1 ± 4.2	<0.0001	6.7 ± 3.9	6.6 ± 2.8	0.91
Lym (×10^3^/μL)	1 ± 0.4	1.5 ± 0.7	<0.0001	1.1 ± 0.6	1.3 ± 0.7	0.09
Mon (×10^3^/μL)	0.3 ± 0.5	0.6 ± 0.4	<0.0001	0.3 ± 0.4	0.4 ± 0.5	0.22
Eos (×10^3^/μL)	0 ± 0	0 ± 0	0.99	0 ± 0	0 ± 0	0.99
Bas (×10^3^/μL)	0.0 ± 0.2	0 ± 0	0.32	0 ± 0	0 ± 0	0.99
CRP (mg/L)	87.2 ± 61.3	13.2 ± 26.3	<0.0001	90.2 ± 60.5	33.1 ± 52	<0.0001
Ferritin (ng/mL)	624 ± 460	483 ± 399	0.01	613 ± 528	605 ± 394	0.79
IL6 (pg/mL)	45.9 ± 59.5	17.5 ± 62	0.01	49.1 ± 47.5	90.5 ± 225	0.12
eGFR (mL/min/1.73 m^2^)	86.5 ± 17.1	94.1 ± 14.3	<0.0001	82.7 ± 25.9	92.9 ± 23.5	0.03
D-Dimer (μg/L)	325 ± 292	283 ± 206	0.26	389 ± 500	505 ± 614	0.7
AST (U/L)	39.1 ± 19	34.6 ± 17.1	0.15	34 ± 19.7	38.4 ± 27	0.4
ALT (U/L)	38.2 ± 22.7	64.1 ± 41	<0.0001	39 ± 48.8	61.2 ± 60.1	0.06
LDH (U/L)	332 ± 114	304 ± 108	0.07	336 ± 105	299 ± 103	0.05

Data expressed as mean  ±  standard deviation; PF: PaO_2_/FiO_2_; pCO_2_: blood partial pressure of CO_2_; RBC: red blood cells; Hb: hemoglobin; WBC: white blood cells; PLT: platelets; Neu: neutrophils; Lym: lymphocytes; Mon: monocytes; Eos: eosinophils; Bas: basophil; CRP: C-reactive protein; IL-6: interleukin 6; eGFR: estimated glomerular filtration rate; AST: aspartate transaminase; ALT: alanine transaminase; LDH: lactate dehydrogenase.

**Table 3 viruses-15-02101-t003:** Course of the disease in patients treated with remdesivir plus SoC or SoC alone.

	Remdesivir (57)	Standard of Care (55)	*p*-Value
Escalation to NIV/VMI, n. (%)	15 (25.3)	2 8 (49.1)	<0.05
Days of hospitalization, n.	15.3 ± 8.3	20.1 ± 8.8	<0.05
Admission to ICU, n.	1	2	
Death, n.	0	1	

## Data Availability

The data presented in this study are available on request from the corresponding author. The data are not publicly available due to ethical reasons.

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
