# Peer review of "Use of Remdesivir in Patients Hospitalized for COVID-19 Pneumonia: Effect on the Hypoxic and Inflammatory State"

_viruses, 2023, doi:10.3390/v15102101_

Round 1

Reviewer 1 Report

This manuscript by Libra and co-workers investigates the effect of remdesivir on the hypoxic and inflammatory state in patients with moderate-to-severe COVID-19. Data from patients admitted to Respiratory Care Unit of University Hospital “Policlinico-San Marco”, Catania, Italy, during the first and second waves of the SARS-CoV-2 pandemic, from June 2020 to May 2021, were retrospectively analyzed.

Specifically, this study enrolled 112 patients hospitalized for COVID-19 pneumonia. All patients required low-flow oxygen; among these, 57 were treated with remdesivir plus standard of care while 55 were treated with standard of care alone. At the time of admission, the two groups were identical in terms of demographic data, clinical history and laboratory parameters.

Changes in hypoxemia and inflammatory markers upon admission and after 5 days of treatment, as well as the clinical course of the disease, were analyzed. It was observed that the ratio of arterial oxygen partial pressure to fractional inspired oxygen increased in the remdesivir plus standard of care group and decreased in the standard of care group and that markers of inflammation decreased only in the remdesivir plus standard of care group. The study results demonstrated that remdesivir reduced the need for ventilatory support and the length of hospital stay resulting in a better disease course in moderate to severe COVID-19 patients.

Overall, this study, which is certainly interesting, perhaps has a limitation due to the small sample of patients. Despite this aspect, the study provides further information on the effect of remdesivir in in patients with moderate-severe COVID-19.

The manuscript is well written and understandable to a specialist readership. The organization and structure of the article are satisfactory. The title clearly indicates the focus of the article and the Abstract section efficiently summarizes the contents of the paper. In the “Introduction” the context of the subject area is properly addressed to justify the study and the objective of the manuscript is clearly indicated. “Materials and Methods” are suitable and the statistical analyses are well defined and appropriate to the design. Figures and Tables are well designed and all necessary for understanding of the text. Conclusion provides interpretation of the results in the context of other evidence. The subject is adequate with the overall scope of “Viruses” (Section: SARS-CoV-2 and COVID-19, Special Issue: COVID-19 and Pneumonia 2.0).

I have only some suggestions:

Materials and Methods

Line 81

Please, change remdesivir with “remdesivir plus standard of care group” (study group)

Line 90

Please, change SARSCoV-2 with “SARS-CoV-2”

Results

Line 178 and Line 183

Please, change In the group treated with remdesivir with “In the group treated with remdesivir plus standard of care group”.

Line 188

Please, change in the remdesivir group with “In the group treated with remdesivir plus standard of care group”.

Table 2. 

Please, change Differences in characteristics of COVID-19 patients on day 0 and day 5, after treatment with Remdesivir or standard of care with “Differences in characteristics of COVID-19 patients on day 0 and day 5, after treatment with remdesivir plus standard of care or with standard of care alone” 

Table 3.  

Please, change Course of the disease in patients treated with remdesivir or standard of care with “Course of the disease in patients treated with remdesivir plus standard of care or with standard of care alone”

Figure 1. 

Please, change Changes in P/F and IL-6 from baseline (day 0) to day5 in patients treated with remdesivir or standard of care with “Changes in P/F and IL-6 from baseline (day 0) to day5 in patients treated with remdesivir plus standard of care or with standard of care alone”

Figure 2. 

Please, change Differences in terms of length of stay in remdesivir and standard of care patients. With “Differences in terms of length of stay in in patients treated with remdesivir plus standard of care or with standard of care alone”.

Line 213 and Line 215

Please, change remdesivir group with “remdesivir plus standard of care group”.

 Conflicts of Interest:

Please specify the role of Andrea Marino (line 405?)

Author Response

Materials and Methods

Line 81

Please, change remdesivir with “remdesivir plus standard of care group” (study group)

Reply: We changed the text as you suggested.

Line 90

Please, change SARSCoV-2 with “SARS-CoV-2”

Reply: We fixed the typo.

Results

Line 178 and Line 183

Please, change In the group treated with remdesivir with “In the group treated with remdesivir plus standard of care group”.

Reply: Thank you for the suggestion. We addressed it.

Line 188

Please, change in the remdesivir group with “In the group treated with remdesivir plus standard of care group”.

Reply: We changed the text.

Table 2. 

Please, change Differences in characteristics of COVID-19 patients on day 0 and day 5, after treatment with Remdesivir or standard of care with “Differences in characteristics of COVID-19 patients on day 0 and day 5, after treatment with remdesivir plus standard of care or with standard of care alone” 

Reply: We changes the text as you suggested.

Table 3.  

Please, change Course of the disease in patients treated with remdesivir or standard of care with “Course of the disease in patients treated with remdesivir plus standard of care or with standard of care alone”

Reply: We addressed what you pointed out.

Figure 1. 

Please, change Changes in P/F and IL-6 from baseline (day 0) to day5 in patients treated with remdesivir or standard of care with “Changes in P/F and IL-6 from baseline (day 0) to day5 in patients treated with remdesivir plus standard of care or with standard of care alone”

Reply:….

Figure 2. 

Please, change Differences in terms of length of stay in remdesivir and standard of care patients. With “Differences in terms of length of stay in in patients treated with remdesivir plus standard of care or with standard of care alone”
Reply:…..

Line 213 and Line 215

Please, change remdesivir group with “remdesivir plus standard of care group”.

Reply: We changed what you suggested.

Conflicts of Interest:

Please specify the role of Andrea Marino (line 405?)

Reply: We added what you pointed out.

Reviewer 2 Report

This study aimed to evaluate the effect of remdesivir on the hypoxic and inflammatory state in patients with moderate-to-severe COVID-19. They found that compared to standard care, remdesivir could help improve hypoxia and inflammation, causing a better course of the disease in moderate-to-severe COVID-19. Although this study is interesting, I have several concerns.

1.     The present work was based on the early COVID-19 pandemic. During the study period, there was no oral antiviral and vaccine. Thus, the findings of this study may no be applied to the condition during this stage. Please discuss this issue in the limitation section.

2.     The case number is small, so the generalizability is also limited.

3.     Please state what is the standard of care, such as tocilizumab or systemic corticosteroid, in the study site during study period.

Author Response

This study aimed to evaluate the effect of remdesivir on the hypoxic and inflammatory state in patients with moderate-to-severe COVID-19. They found that compared to standard care, remdesivir could help improve hypoxia and inflammation, causing a better course of the disease in moderate-to-severe COVID-19. Although this study is interesting, I have several concerns.

  1. The present work was based on the early COVID-19 pandemic. During the study period, there was no oral antiviral and vaccine. Thus, the findings of this study may no be applied to the condition during this stage. Please discuss this issue in the limitation section.

Reply: Thank you for the valuable suggestions. We added what you highlighted.

  1. The case number is small, so the generalizability is also limited.

Reply: We also remarked this limitation

  1. Please state what is the standard of care, such as tocilizumab or systemic corticosteroid, in the study site during study period.

Reply: We better clarified that in the method section.